# Factors contributing to low tuberculosis diagnosis among children aged 0–14 years in Gem Sub County in Siaya County, Kenya

**Lilian Atieno Okumu**[1]*, **Daniel Ogungu Onguru**[1], **David Otieno Odongo**[2], **James Onyuro Oketch**[1]

**1** School of Health Sciences, Jaramogi Oginga Odinga University of Science and Technology, Bondo, Kenya, **2** Affiliate Faculty, Foundation of Healthcare Technologies Society, FHTS, New Delhi, India

* okumulilian@yahoo.com

## Abstract

Tuberculosis (TB) is a top global health challenge, with 1.2 million children aged 0–14 years becoming ill with TB every year. Yet, a significant proportion remain undiagnosed or missed due to diagnostic barriers. This cross-sectional study employing an embedded mixed method approach investigated factors contributing to low tuberculosis diagnosis among children in Gem Sub County, Siaya County Kenya, a high burden region. Data was collected from 71 healthcare workers (HCWs) and 16 respondents across six wards using semi-structured questionnaires and TB register abstraction. Qualitative data underwent deductive thematic analysis while quantitative data was analyzed using descriptive statistics and logistic regression in SPSS version 27. About a third (31.3%) of TB cases in children required three or more facility visits before diagnosis, with some needing up to 12 visits. Costly and inaccessible chest X-rays and GeneXpert underutilization due to difficulties in sputum and alternative sample (gastric lavage, nasopharyngeal aspirates) collection were key diagnostic constraints. Delayed care-seeking due to stigma, misconceptions (56.3% linked childhood TB to HIV) and low TB symptom awareness (37.5% of children presented with ≤2 symptoms) were patient related factors associated with low TB diagnosis. About 56.3% of caregivers sought routine TB screening, but 62.5% sought care only after persistent symptoms. Clinical officers showed substantially higher odds of reporting confidence in sample collection (OR=34, 95% CI 3.81–303.21, p=0.002) and GeneXpert interpretation (OR=15, 95% CI 3.20–70.39, p<0.001) than nurses. Trained HCWs were 7.16 times more likely to interpret X-rays accurately (95% CI 2.16–23.67, p<0.001). Enhanced HCW training, improved diagnostic access, targeted community education on stigma and misconceptions are critical for early TB detection in children.

**Data availability statement:** All relevant data are within the manuscript.

**Funding:** The author(s) received no specific funding for this work.

**Competing interests:** The authors have declared that no competing interests exist.

## Introduction

Tuberculosis (TB), caused by *Mycobacterium Tuberculosis*, transmitted through respiratory droplets from infected individuals, is a formidable global health challenge. In 2023, an estimated 10.8 million new cases occurred worldwide, including 1.3 million among children aged 0–14 years, equivalent to 12% of the total TB burden [1]. Of these, 8.2 million people were reported as newly diagnosed, up from 7.5 million in 2022 and 7.1 million in 2019 [1]. Furthermore, despite being preventable and treatable [2], TB causes deaths, claiming approximately 1.3 million lives annually [1]. Yet, many cases remain undiagnosed or underreported, in particular, pediatric population [3]. Previous research has shown that this is mainly attributed to nonspecific symptoms, low bacillary loads, and challenges in sputum sample collection in children, and limited access to diagnostic tools in high-burden settings [4,5].

Kenya, ranking fourth in Africa for TB burden after Ethiopia, Nigeria, and South Africa, is among the top 10 high-burden countries globally. The Kenya National TB prevalence survey of 2015/2016, reported a TB prevalence of 558 per 100,000 population with nearly half of the cases missed annually [6]. Notably, 21% of the patients with TB symptoms sought prior care at private clinics and chemists [6]. TB is often under-reported, with children representing only 8% of the 2,903 TB cases reported in Siaya County in 2021 according to Kenya TB surveillance data 2021, below the national target of 12%. In Gem Sub-County, only 4% of the 619 TB cases reported in 2021 were pediatric, well below the expected 10–15% range, highlighting significant diagnostic gaps.

Multiple factors drive TB under-diagnosis in childhood population. This includes the absence of clear clinical features, diagnostic tool limitations, and patient- and facility-related barriers [7,8]. A systematic review by Divala et al. (2022) demonstrated that TB symptom-based screening is inconsistently implemented [7]. Patients attending health facilities are not consistently asked about respiratory symptoms nor promptly offered diagnostic tests once the symptoms are identified. Under-reporting and misdiagnosis contribute to delayed treatment, ongoing community transmission and risk of developing drug-resistant TB. Hence, the accurate and timely identification of TB in pediatric populations is critical for effective treatment and preventing community transmission [5].

Despite these challenges, local data on the nature and drivers of low TB diagnosis in children remains limited. This is concerning given the burden of TB in Kenya. The reliability of GeneXpert in children, limited by low bacillary loads, remains under scrutiny, necessitating further investigation, to address the rising burden [5,8]. The purpose of this study was to evaluate the constraints to current Tuberculosis diagnostic approaches in children, with a focus on patient related and facility-based barriers in Gem Sub-County, Siaya County, Kenya. By identifying these barriers, the findings aim to inform clinical practices to improve early detection, reduce missed cases, and support TB control efforts in Siaya County and similar geographic areas.

## Methodology

### Study area and design

This cross-sectional study conducted in Gem sub-county, Siaya County, Kenya employed an embedded mixed-methods design, with two strands (quantitative and qualitative component) mixed at design phase. Quantitative data was the primary component and qualitative data as a supportive secondary component to elicit richer explanations of TB diagnostic constraints and experiences. This approach allowed efficient concurrent collection of both data types from HCWs in a single survey instrument/tool. Data from caregivers/children (n = 16) were collected via separate semi-structured open-ended interviews using a similar approach. Participants were recruited between 13th December 2024 and 16th January 2025. Siaya County is one of the 10 highest burden TB Counties in Kenya. The county consists of six sub-counties [9]. The study specifically focused on Gem sub-county, which has one of the highest tuberculosis notification rates, as reported in the Kenya National tuberculosis, leprosy, and lung disease 2022 annual report. As per Kenya master health facility registry (KMHFR), Gem subcounty is served by four levels of healthcare facilities, Level 1 (community health services), Level 2 (dispensaries), Level 3 (health centers), and Level 4 (sub-county hospitals).

### Study population

The study population comprised healthcare workers, parents/ caregivers/ children aged 0–14 years diagnosed with TB at health facilities in Gem Siaya, and receiving treatment during the study period, provided they or their caregivers gave written informed consent or assent. Healthcare workers (HCWs), including doctors, nurses, clinical officers, laboratory technologists, and pharmaceutical technologists, employed for at least two months with pediatric TB experience and consented were included. Exclusions included HCWs with less than two months' employment, or lacking pediatric TB experience, and children transferred from other facilities.

### Sampling technique

A multistage sampling approach was used to select study participants. First, twenty TB diagnostic facilities were purposively selected based on their high TB caseloads. These facilities included two sub-county hospitals, and a mix of health centers, dispensaries, and private/mission facilities. Out of these, only seven [7] facilities had active childhood TB cases in the TIBU register, yielding a total of 17 children receiving TB treatment. Of these, 16 cases were included and one case was excluded because of transfer to another county and inability to trace via phone. Third, for healthcare workers (HCWs), stratified sampling was initially used to ensure representation across different HCW cadres. This led to 71 HCWs with substantial experience in pediatric TB management being purposively selected.

### Sample size calculation

Sample size was determined using the Taro Yamane formula (Yamane, 1967), yielding 76 HCWs and 16 patients/ caregivers.

The formula is defined as:

$$n = \frac{N}{1 + N(e)^2}$$

where:

n = desired sample size

N = population size

e = acceptable margin of error

For the number of caregivers/patients, the sample size is calculated as:

$$n = \frac{17}{1 + 17 * (0.05)^2} 16.30 \approx 16$$

For the HCWs, the sample size is calculated as;

$$n = \frac{86}{1 + 86 * (0.05)^2} = 70.781 \approx 71$$

This implied that approximately 71 HCWs and 16 patients/caregivers or guardians were sampled from the selected health facilities to achieve a 95% confidence level with a 5% margin of error, assuming a response distribution of 50%.

## Data collection procedure

Data was collected using validated semi structured questionnaires, using the Kobo Collect platform on Android tablets. Both quantitative and qualitative questions in form of addons were collected from all the respondents. The questionnaires captured knowledge of TB, diagnostic practices, and narrative insights on challenges encountered in diagnosing TB in children. TB4 register (facility level register) abstraction provided case data in selected facilities. The data collection tools were pretested to ensure reliability, and daily debriefings with research assistants ensured data quality.

## Data management and analysis

Data was exported from Kobo collect to Microsoft Excel, for cleaning and initial review, then coded, and imported into SPSS version 27 for analysis. Descriptive statistics was used to summarize participant characteristics and diagnostic practices. Bivariate analysis was used to determine associations between independent variables (HCW training, cadre) and specific outcomes related to low TB diagnosis; (1) accurate chest X-ray interpretation (defined as correctly identifying TB-related abnormalities per WHO guidelines), (2) confidence in sample collection (self-reported ability to collect sputum or alternative samples), and (3) confidence in GeneXpert interpretation (self-reported ability to interpret GeneXpert results). Univariate binary logistic regression was used to calculate crude odds ratios (ORs) for each predictor-outcome pair separately to explore associations, with a p-value $< 0.05$ considered statistically significant. Due to multiple distinct outcomes and a modest sample size (71 HCWs, 16 patients/caregivers), multivariate analysis was not performed to avoid overfitting, and no correction for multiple comparisons was applied given the exploratory nature of the study.

Qualitative data from responses to open-ended questions embedded in the questionnaires, including detailed notes taken by trained research assistants during the sessions were compiled, organized, and anonymized to ensure confidentiality. Analysis was conducted manually in Microsoft Word using deductive thematic analysis guided by the study objectives, with flexibility for inductive coding of emergent themes. The process followed four steps: (1) familiarization through repeated reading of all notes and responses; (2) initial open coding to identify concepts; (3) axial coding to group related codes into categories; and (4) development of overarching themes through iterative discussion and review by the research team.

## Ethical considerations

Authorization was sought from the Jaramogi Oginga Odinga University of Science and Technology, Board of Postgraduate studies. Ethical approval from Institutional Scientific Ethics Review Committee of University of East Africa Baraton (UEAB/ISERC/05/11/2024), thereafter research permit from the National Commission for Science, Technology and Innovations

(NACOSTI P/24/42206) before the commencement of the study. Consent was obtained from all participants. Informed written consent forms were provided in English, Kiswahili and Luo to adult participants and parents/guardians of children prior to participation, with assent obtained for children. In particular, to cater for the children, after written consent from parents/guardians, assent was sought before proceeding to the survey. The consent included purpose of the study, risk and benefits, voluntary participation, confidentiality and rights of the participants. Data was anonymized and stored in access-controlled computers with access limited to the research team. No identifiable information was used in any study report, presentations or publications.

## Results

### Sociodemographic characteristics of the respondents

The study involved 71 healthcare workers and 16 respondents (13 parents/guardians and 3 children). About half of health-care workers were female (38, 53.5%), while over half of other respondents (9, 56.3%) were aged 35 years and above. Most HCWs (44, 62%) were aged 21–34 years, and majority (55, 77.5%) had up to diploma training, 11(15.5%) had a degree, and a smaller proportion (5, 7%) had only secondary education Table 1.

### Diagnostic constraints for paediatric tuberculosis

**Chest X-ray constraints.** High costs and limited access to chest X-ray facilities were reported barriers. A 34-year-old nurse stated, *"Most patients cannot afford X-ray charges, hindering proper diagnosis."* A 29-year-old clinical officer pointed out, *"one would use over two thousand Kenya shillings (KSH 2000) just to get the X-ray done, while at home they lack food."* Only 23.9% of HCWs could interpret pediatric radiographs, citing poor image quality and inadequate training. A 40-year-old laboratory technologist noted, *"X-ray interpretation requires highly trained personnel, often unavailable in remote settings."* Additionally, a 27-year-old CO warned, *"wrong positioning of the patient could lead to wrong interpretation and hence incorrect diagnosis."* Radiation concerns and transportation costs further limited use. A 40-year-old CO reported, *"Facilities equipped with X-ray equipment are quite a distant from patients reach."* making referrals a challenge.

**Gene X-pert/smear microscopy challenges.** Sample collection challenges were reported in young children unable to expectorate sputum. Alternative methods (gastric lavage, nasopharyngeal aspirates) required specialized skills, lacking in many HCWs. Stock-outs of GeneXpert cartridges, nasogastric tubes, and reagents disrupted diagnosis. A 26-year-old link assistant noted, *"Obtaining samples from infants and children who are not able to expectorate is a challenge due to inadequate trained personnel in nasopharyngeal aspiration."* A 45-year-old CO noted, *"Stock-outs of NG tubes and essential supplies disrupt sample collection."* Additionally, a 32-year-old CO noted, *"Transportation of the samples from facility A to the laboratory hubs interferes with sample quality,"* while a 40-year-old Nursing officer emphasized high costs. *"It is expensive to transport the sample to a definitive facility."*

**WHO diagnostic algorithm constraints.** Underuse of TB algorithm was attributed to complexity and lack of training (69% of HCWs lacked recent pediatric TB training). Algorithms were not available at all service delivery points, and frequent updates without dissemination led to misinterpretation. A 48-year-old nurse reported, *"Misunderstanding of algorithms requires regular sensitizations."* A 29-year-old CO noted, *"Algorithms keep changing especially for management of pediatrics on TB treatment."* A 44-yearold clinician stated, *"Not all service delivery points (SDPs) have access to the algorithms."* (Table 2).

**Patient related factors contributing to low tuberculosis diagnosis in children.** Delayed care-seeking was common due to low symptom awareness (37.5% of children had ≤2 symptoms). A 62-year-old woman admitted, *"I did not know the signs and symptoms of TB let alone in children,"* A 27-year-old woman initially dismissed symptoms in a child because she believed TB only affected older people. *"I assumed TB was a disease of the old,"* she confessed. Further, financial constraints and long distances to facilities were barriers. A 32-year-old woman expressed her struggles: *"No

**Table 1. Sociodemographic Characteristics of the Respondents.**

**Healthcare workers**

| Variable | Category | Frequency (n=71) | Percentage |
|---|---|---|---|
| Gender | Female | 38 | 53.5 |
| | Male | 33 | 46.5 |
| Age Category | 21-34 years | 44 | 38 |
| | 35+ years | 27 | 62 |
| Education level | Secondary | 5 | 7 |
| | Diploma | 55 | 77.5 |
| | Degree | 11 | 15.5 |
| Cadre | Clinical Officer | 18 | 25.4 |
| | Nurse | 24 | 33.8 |
| | Laboratory Technologist | 15 | 21.1 |
| | Peer educator | 4 | 5.6 |
| | AYP | 3 | 4.2 |
| | HTS provider | 2 | 2.8 |
| | Nutritionist | 2 | 2.8 |
| | Pharmaceutical Technologist | 1 | 1.4 |
| | Link assistant | 1 | 1.4 |
| | TB champion | 1 | 1.4 |
| | SCTLC | 1 | 1.4 |
| Work Experience | < 2 years | 13 | 18.3 |
| | 3-5 years | 16 | 22.5 |
| | 6-9 years | 21 | 29.6 |
| | 10+ years | 21 | 29.6 |
| Work duration in current facility | 5+ years | 16 | 22.5 |
| | 1-4 years | 55 | 77.5 |
| Facility | Dispensary | 7 | 9.9 |
| | Health Centre | 48 | 67.6 |
| | Subcounty Hospital | 16 | 22.5 |
| Current Employer | MOH | 34 | 47.9 |
| | MOH/Partner support | 20 | 28.2 |
| | Faith based | 7 | 9.9 |
| | Private | 10 | 14.1 |

**Clients**

| Variable | Category | Frequency (n=16) | Percentage |
|---|---|---|---|
| Person interviewed | Client | 3 | 18.8 |
| | Parent/guardian | 13 | 81.3 |
| Gender | Female | 13 | 81.3 |
| | Male | 3 | 18.8 |
| Age Category | 12-18 | 3 | 18.8 |
| | 19-34 | 4 | 25 |
| | 35+ | 9 | 56.3 |
| Education level | Primary | 10 | 62.5 |
| | Secondary | 4 | 25 |
| | College | 2 | 12.5 |

**Table 2. Summary of Key Insights on Constraints to Tuberculosis Diagnosis.**

| Diagnostic Approach | Emerging Themes | Key Insights | Representative Quotes |
|---|---|---|---|
| Chest X-Ray | Cost and Accessibility Constraints | High costs and distance to facilities hinder early detection | "One would use over 2000/= just to get the X-ray done, while at home they lack food." (29-year-old CO) |
| | Image Quality & Interpretation Issues | Poor image quality and lack of trained personnel lead to misdiagnosis. | "Wrong positioning of the patient could lead to wrong interpretation and hence incorrect diagnosis." (27year-old CO) |
| GeneXpert/ Smear Microscopy | Sample Collection Difficulties | Young children cannot expectorate sputum, making sample collection difficult. | "Gastric aspirate collection in under-fives is not easy as it requires much courage." (Nursing Officer) |
| WHO Diagnostic Algorithms | Lack of Training and Awareness on Updates | Frequent updates without dissemination result in misinterpretation and poor implementation. | "Misunderstanding of the algorithms hence need for regular sensitizations." (48-year-old Nurse) |
| | Misinterpretation and Complexity of Algorithms | Some healthcare workers find the algorithms difficult to interpret and apply. | "Misinterpretations are common." (30-year-old Laboratory Technologist) |

*money and long distance in accessing health service makes it difficult."* Similarly, a 38-year-old and a 14-year-old girl struggled with reaching healthcare facilities in time.

Stigma and misconceptions (56.3% linked TB to HIV, 37.5% believed TB was hereditary or a curse) delayed diagnosis, with fears of community judgment. A 49-year-old woman hesitated taking her child for TB diagnosis due to stigma. She admitted, *" Stigma of being diagnosed with TB. Fear of the community's perception of those who have TB could be having HIV/AIDs delayed my decision."* A 37-year-old mother couldn't believe a young child could have TB. *"I didn't think such a young child could have TB."* For a 12-year-old girl, the possibility of TB was only considered only after further testing. *"Nobody thought it was TB until the doctor suggested I go for an X-ray. It could be TB,"* she recalled

Quantitative findings confirmed the respondents' narratives. Over half (56.3%) of caregivers sought routine TB screening, but 62.5% only sought care after persistent symptoms. Most respondents (14, 87.5%) knew at least three TB signs and symptoms. Over a third (6, 37.5%) of children presented with only 0–2 symptoms before diagnosis. Notably, slightly over half (9, 56.3%) associated TB with HIV. (Table 3).

Cough of any duration was the most common symptom (87.5%), and most children (75%) presented with it prior to diagnosis. Chest pain was identified by 87.5% of respondents but was present in only 56.3% of children before diagnosis. Weight loss was acknowledged by 87.5% of respondents, but only 62.5% of children presented with it. Fever was mentioned by 62.5% of respondents, yet only 37.5% of children experienced it before diagnosis. Night sweats were identified by 56.3% of respondents but present in only 43.8% of children (Fig 1).

**Facility related factors influencing tuberculosis diagnosis in children.** Only 31% of HCWs had recent pediatric TB training, with 26 (53.1%) citing lack of opportunity or unawareness of such training (12, 24.5%). Most HCWs (65, 91.5%) could correctly identify three or more signs of active TB, but only 17(23.9%) could interpret pediatric radiographs. Most respondents (48, 67.6%) believed that all HCWs, regardless of cadre, should receive training on childhood TB management. Almost all respondents (69, 97.2%) were aware that a child can be infected with TB more than once in their lifetime. Table 4

Multiple facility visits were required before diagnosis (Fig 2). A 42-year-old parent noted, *"We sought help from different big hospitals within the County but in vain, the 6th visit is what came to our rescue"* A 30-year-old woman echoed this saying, *"The medics failed to get the right diagnosis on time."* Similarly, a 12-year-old girl was only diagnosed after further testing. *"Nobody thought it was TB until the doctor suggested I go for an X-ray. It could be TB."* A 51-year-old man

**Table 3. Patient Related Factors Contributing to Delayed TB Diagnosis.**

| Variable | Frequency | Percentage |
|---|---|---|
| **Know signs and symptoms of TB** | | |
| No | 2 | 12.5 |
| Yes | 14 | 87.5 |
| **Number of TB known signs and symptoms** | | |
| 0-2 | 2 | 12.5 |
| 3+ | 14 | 87.5 |
| **Number of signs and symptoms child presented with before diagnosis** | | |
| 0-2 | 6 | 37.5 |
| 3+ | 10 | 62.5 |
| **Myths about Tuberculosis disease** | | |
| TB is a curse | 6 | 37.5 |
| TB is not curable | 2 | 12.5 |
| TB is hereditary | 6 | 37.5 |
| If you have TB then you have HIV | 9 | 56.3 |
| TB is for people who drink alcohol | 1 | 6.3 |
| **Take children under your care to hospital for TB screening** | | |
| No | 7 | 43.8 |
| Yes | 9 | 56.3 |
| **Point at which you take children to hospital** | | |
| To perform early diagnosis of TB | 4 | 44.4 |
| Routine check for TB disease | 3 | 33.3 |
| Do not undergo early screening until signs and symptoms manifest | 2 | 22.2 |
| **Where children are taken for TB screening** | | |
| Public hospital | 9 | 69.2 |
| Private hospital | 3 | 23.1 |
| Community level (CHP) | 1 | 7.7 |
| **Where I seek help when a child falls sick** | | |
| Buy medicine at the chemist/pharmacy | 4 | 25 |
| Go to health facility | 12 | 75 |
| **Point at which child is taken to health facility after onset of illness** | | |
| Immediately he or she falls sick | 6 | 37.5 |
| When symptoms persist | 10 | 62.5 |

expressed frustration after numerous visits without receiving proper guidance. *"There was no proper medication all this time. I was visiting different facilities with my child. No one ever mentioned to me about required tests."* This was confirmed by HCWs noting that TB symptoms in children are not easily identifiable due to similarity with other diseases. *"Most children diagnosed don't present with easily identifiable symptoms,"* noted a 48-year-old Nurse. A 27-year-old Laboratory Technologist reiterated, *"The similarity of TB signs and symptoms to other diseases makes clinical diagnosis hard.* Notably one HCW, a 33-year-old link assistant noted, *"Chest X-rays are not advisable in children due to harmful rays, which may lead to missed cases due to fear."* Additionally, the HCWs identified persistent shortage of adequately trained personnel. A 40-year-old registered nurse stated, *"Lack of enough trained personnel, and understaffing make TB diagnosis difficult."*

Clinical officers showed substantially higher odds of confidence in sample collection (OR 34, 95% CI 3.81–303.21, p = 0.002) and GeneXpert interpretation (OR 15, 95% CI 3.20–70.39, p < 0.001) than nurses. Healthcare workers who had attended a training workshop on pediatric Tuberculosis were 7.1 times more likely to interpret chest X ray than those who

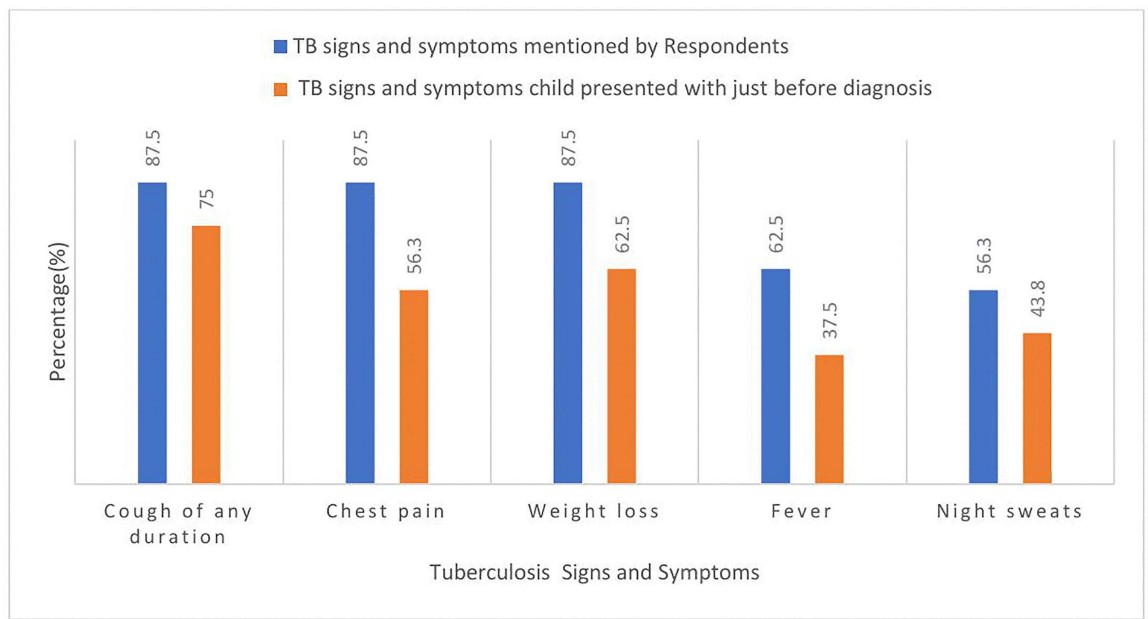

**Fig 1. Prevalence of common Tuberculosis signs and symptoms mentioned by caregivers compared to those the children presented with at diagnosis.**

had not (OR 7.16, 95% CI (2.16–23.67), p < 0.001). However, knowledge of major signs and symptoms of TB disease, perception on who should be trained on childhood TB diagnosis were not statistically significant (Table 5).

## Discussion

### Introduction

Childhood tuberculosis (TB) remains significantly underdiagnosed in high-burden settings. This is attributed to challenges in bacteriological confirmation, non-specific clinical presentation, and health system barriers that delay or prevent timely detection [3,5]. To this end, interventions targeting these barriers have shown promise in improving case detection at primary care levels [8,10,11]. In this present study, in Gem Sub-County, Siaya County, Kenya, we explored diagnostic constraints for Tuberculosis in children aged 0–14 years from the interviews with healthcare workers (HCWs) and caregivers/children, complemented by health facility data. Here in, we demonstrate that the findings align with and extend this prior evidence by documenting local insights of ongoing TB diagnosis challenges in a rural setup in Kenya.

### Diagnostic constraints for tuberculosis in children

The study revealed that limited access to and high costs of chest X-rays, radiation risks, combined with low HCW proficiency in interpreting pediatric radiographs (only 23.9% reported competence) contributed to diagnostic delay. Qualitative accounts emphasized financial, access, and training barriers and transportation difficulties. These findings are consistent with reports from resource-constrained settings, where financial barriers including transportation costs hinder timely TB diagnosis [9]. TB patients in resource-constrained settings, often incur unmanageable costs while seeking and staying in care for the full duration of anti-TB treatment. Furthermore, the findings complement a previous study which indicated current diagnosis as suboptimal due to challenges in sample collection, and inadequate access to diagnostic tools in healthcare settings in high burden areas [5]. A previous study in six high TB burden settings, including Ivory coast, Cameroon,

**Table 4. Healthcare Worker Related Factors on Childhood TB Diagnosis.**

| Variable | Frequency | Percentage |
|---|---|---|
| **Attended training/workshop on Paediatric TB in the past 12 months** | | |
| No | 49 | 69 |
| Yes | 22 | 31 |
| **If No Why** | | |
| Was committed to other duties | 9 | 18.4 |
| Not aware that the above training exists | 12 | 24.5 |
| Not invited/provided an opportunity | 26 | 53.1 |
| There has been no training specific to paediatric TB | 2 | 4.1 |
| **Who should be trained on childhood TB Management** | | |
| Specific group of HCWs | 23 | 32.4 |
| All HCWs regardless of cadre | 48 | 67.6 |
| **Number of TB Signs and symptoms mentioned by respondents** | | |
| 0-2 signs | 6 | 8.5 |
| 3+signs | 65 | 91.5 |
| **A child can be infected with TB more than once in their lifetime** | | |
| Don't know | 2 | 2.8 |
| Yes | 69 | 97.2 |
| **Situations when Patients are given health education messages about TB** | | |
| During World TB Day | 61 | 86 |
| During BCG Immunization | 55 | 77.4 |
| During General health education delivered in clinical settings | 69 | 97.1 |
| With suspected or confirmed cases only (i.e., no family members) | 5 | 7.0 |
| With suspected cases and their families in a clinical setting | 58 | 81.7 |
| With confirmed patients and their families | 60 | 84.5 |
| Health education on TB is generally not done with patients | 1 | 1.4 |
| **Primary diagnostic test to confirm or rule out active TB in children** | | |
| GeneXpert | 35 | 49.3 |
| Chest X Ray | 10 | 14.1 |
| Sputum smear microscopy | 15 | 21.2 |
| TB LAM | 11 | 15.5 |
| **Ever collected sample for TB diagnosis** | | |
| No | 30 | 42.3 |
| Yes | 41 | 57.7 |
| **If Yes type of sample collected** | | |
| Sputum | 17 | 41.5 |
| Gastric aspirate | 15 | 36.6 |
| Nasopharyngeal aspirate | 9 | 22 |
| **How confident are you in collecting sample from Children** | | |
| Very confident | 16 | 22.5 |
| Confident enough | 26 | 36.6 |
| Not so sure | 29 | 40.8 |
| **Confidence in interpreting microscopy, GeneXpert TB results in children** | | |
| No | 32 | 45.1 |
| Yes | 39 | 54.9 |
| **Able to interpret radiography images in children** | | |
| No | 54 | 76.1 |

*(Continued)*

**Table 4.** (Continued)

| Variable | Frequency | Percentage |
|---|---|---|
| Yes | 17 | 23.9 |
| **Ever diagnosed TB disease in a child** | | |
| No | 34 | 47.9 |
| Yes | 37 | 52.1 |
| **Method used to diagnose TB in a child** | | |
| Clinical diagnosis (algorithm guide by WHO) | 24 | 33.8 |
| GeneXpert test | 20 | 28.2 |
| Smear microscopy | 8 | 11.3 |
| Other | 1 | 1.4 |
| **Skill/ training level needed to effectively conduct DOT in TB patient** | | |
| Any healthcare worker regardless of clinical training | 33 | 46.5 |
| Any healthcare worker with clinical training | 22 | 31 |
| This is not a technical activity that requires training | 15 | 21.1 |
| Only highly qualified trained healthcare workers | 1 | 1.4 |

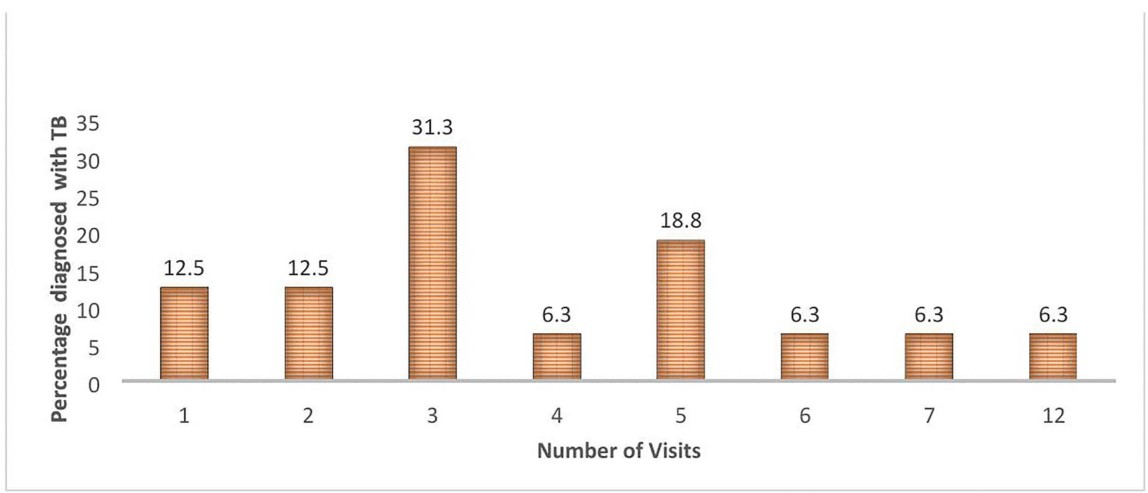

**Fig 2. Number of health facility visits before Tuberculosis diagnosis.**

Cambodia, Uganda among other countries demonstrated that improved access to radiography can significantly enhance TB detection rates in children [12].

Stock-outs of GeneXpert cartridges and reagents observed in our study echo global reports of supply chain disruptions in low-resource settings. In addition to supply chain constraints, we observed underutilization of WHO algorithms attributed to perceived complexity and inadequate training of HCWs. These findings are consistent with evidence from Zambia, where implementation of multicomponent strategy, including capacity building resulted in higher childhood case notifications rates [13]. Likewise, another study has emphasized the need for simplified, disseminated algorithms to address the poor integration of TB screening into pediatric care [11]. Together, diagnosis of TB in children relies on thorough assessment of all the evidence derived from a careful history of exposure, clinical examination and relevant

**Table 5. Logistic Regression Analysis of Facility/Healthcare worker Factors Associated with Childhood TB Diagnosis.**

| Variable | Odds Ratio | 95% CI | p-value |
|---|---|---|---|
| **Confidence in TB sample collection from children** | | | |
| Nursing Officer | Ref | | |
| Clinical Officer | 34 | 3.81–303.22 | **0.002** |
| Laboratory Technologist | 4 | 1.02–15.72 | **0.047** |
| **Ever collected sample for TB diagnosis from children** | | | |
| Nursing Officer | Ref | | |
| Clinical Officer | 34 | 3.81–303.22 | **0.002** |
| Laboratory Technologist | 4 | 1.02–15.72 | **0.047** |
| **Confidence in interpreting microscopy/GeneXpert results** | | | |
| Nursing Officer | Ref | | |
| Clinical Officer | 15 | 3.20–70.39 | **0.001** |
| **Attended TB training workshop in past 12 months** | | | |
| Nursing Officer (reference) | Ref | | |
| Clinical Officer | 10 | 2.34–42.78 | **0.002** |
| Laboratory Technologist | 0.36 | 0.04–3.55 | 0.379 |
| **Ability to interpret pediatric chest radiography** | | | |
| Nursing Officer | Ref | | |
| Clinical Officer | 55 | 8.18–369.85 | **<0.001** |
| **Ever diagnosed TB disease in a child** | | | |
| Nursing Officer | Ref | | |
| Clinical Officer | 7 | 1.59–30.80 | **0.01** |
| Laboratory Technologist | 1.6 | 0.44–5.87 | 0.478 |
| **Frequently sensitizes patients on active TB** | | | |
| Nursing Officer | Ref | | |
| Clinical Officer | 0.74 | 0.04–12.67 | 0.835 |
| Laboratory Technologist | 0.17 | 0.02–1.86 | 0.148 |
| **Who should be trained on childhood TB** | | | |
| Nursing Officer | Ref | | |
| Clinical Officer | 0.52 | 0.14–1.97 | 0.338 |
| Laboratory Technologist | 0.92 | 0.21–3.99 | 0.908 |
| **Training workshop attendance on confidence in CXR interpretation** | | | |
| No | Ref | | |
| Yes | 7.16 | 2.16-23.67 | **0.001** |
| **Training workshop attendance on confidence in sample collection** | | | |
| No | Ref | | |
| Yes | 7.15 | 1.87-27.36 | **0.004** |

investigations. Although bacteriological confirmation of TB is not always feasible, it remains a critical component and should be sought whenever feasible by microscopy, culture or WHO-endorsed genotypic (molecular) testing of respiratory or non-respiratory samples using the GeneXpert to provide diagnostic support.

## Patient-related factors affecting tuberculosis diagnosis in children

Low symptom recognition among caregivers and older children, coupled with misconceptions (56.3% associating TB with HIV) and stigma, contributed substantially to delayed care-seeking and prolonged diagnostic journeys. Many caregivers

and even older children failed to recognize symptoms suggestive of TB, delaying their decision to seek medical attention. These findings are consistent with the 2016 Kenya TB Prevalence Survey, where 64.9% of individuals with TB symptoms had not sought any health care prior to the survey [6]. Similarly, a systematic review demonstrated that reliance on symptoms-based screening alone is a main contributor to underdiagnosis of TB [7]. Across studies included in the review, the proportion of symptomatic individuals who were offered diagnostic investigations was low, with a median of 38% (IQR 14–44; range 4–84) [7]. A predominantly reactive healthcare seeking approach disadvantages early TB detection and treatment, leading to significant underreporting or missed cases [3,8]. The WHO Report 2021, posits that TB detection in children is particularly disadvantaged due to the paucibacillary nature of pediatric TB, the overlap of nonspecific symptoms with other childhood illnesses, and that most children do not excrete enough bacilli to be detectable by available bacteriological tests (WHO Report 2021, Module 5). These barriers when combined with delayed care seeking impede early TB detection and treatment initiation.

Misconceptions that TB primarily affects adults, along with persistent stigma, contributed to symptom neglect in pediatric cases. Stigma, particularly TB's association with HIV, has been consistently documented as a major barrier to timely diagnosis across high-burden settings [14,15]. Moreover, the repeated missed opportunities observed across multiple visits demonstrate the need for strict adherence to standardized protocols. Evidence from review studies indicate that non-adherence to guidelines is a key factor driving missed diagnoses [7,8]. Improved training and mentorship of healthcare providers on standardized diagnostic protocols alongside community level education may improve TB detection and outcomes in pediatric population.

### Facility/ healthcare worker factors affecting tuberculosis diagnosis in children

The study revealed low training uptake (31% of HCWs trained) and cadre differences in diagnostic confidence (clinical officers being more confident). Reasons cited ranged from lack of invitations or being provided a chance to being unaware of training opportunities. This is supported by qualitative findings of training gaps and resource shortages. Non-awareness suggests gaps in information dissemination rather than a lack of willingness to participate in training. Qualitative insights revealed skill deficits in pediatric sample collection for GeneXpert testing, in particular, nasopharyngeal aspirates and gastric lavage sample collection. These specimens have highest bacterial yield [5], but require specialized skills in collection except for stool sample collection. Furthermore, healthcare workers reported that TB symptoms in children are not easily identifiable leading to many cases remaining misdiagnosed, not-diagnosed, or untreated. Similar to our findings, the non-specific nature of TB symptoms in children has been shown as a major diagnostic challenge identified in previous studies [5,11].

Available evidence demonstrates that structured training programs significantly improve TB detection rates by enhancing HCWs' knowledge and diagnostic confidence [5]. A study in Ghana found that undertrained HCWs were less likely to adhere to national TB diagnostic protocols, were managing TB for other health conditions, leading to delay in case identification [16]. As observed in our study and other studies, HCWs with more extensive medical training, such as general practitioners, demonstrate better TB diagnostic skills due to their participation in workshops and professional exchanges [17]. On the other hand, a previous study indicated that HCWs improved interpretation skills in diagnosing paediatric TB after attending a CXR training course [18]. This is because targeted training programs that address the specific needs of different HCW groups offers the advantage of gaining the required skills to support childhood TB diagnosis. Such training could help to address misconceptions from some healthcare workers that Chest X-rays are not advisable in children due to harmful rays. Although the radiation dose from a single pediatric chest X-ray is very low and considered safe, evidence shows that the benefits of accurate TB diagnosis far outweigh the minimal risks [12].

Shortages of diagnostic tools and stock-outs are consistent with global challenges in TB case detection [1]. Given that GeneXpert is the preferred diagnostic test for childhood TB, it is concerning that only 49.3% of HCWs identified it as the primary test. Additionally, difficulty in obtaining quality sputum samples from children disadvantaged TB diagnosis,

necessitating alternative diagnostic methods such as gastric aspirates. Yet, a few HCWs had confidence in collecting alternative samples, in particular gastric lavage. The absence of essential tools, coupled with poor sample collection techniques among healthcare workers, was a significant constraint to TB diagnosis. Accordingly, for children with TB symptoms in settings without resources, an algorithmic approach may guide the initiation of TB treatment, when imaging or microbiological testing are out of reach [11]. Our findings confirm that diagnostic delays in pediatric TB also stem from systemic deficiencies in medical infrastructure and resource availability.

The study had strengths and limitations. This study benefitted from a mixed methods approach effectively capturing the diagnostic challenges faced in childhood TB detection. The study only included children or parents/guardians whose children had a confirmed TB disease to avoid misclassification bias in the analysis of associated factors. However, this selection may limit generalizability to undiagnosed or presumptive cases. The small sample size, particularly for caregivers/parents/children (n = 16), was constrained by the low number of confirmed and active childhood TB cases in the study area, possibly due to underdiagnosis. Further, the wide confidence intervals in HCW cadre differences necessitate cautious interpretation. The reported associations should be viewed as preliminary signals of cadre-specific differences rather than definitive evidence of effect size. The study relied on self-reported data from healthcare workers which may have introduced biases. In particular, healthcare workers might have provided desirable responses regarding their TB diagnostic practices.

In conclusion, although many barriers to Tuberculosis diagnosis have been described previously, this study provides several important new insights. Pediatric TB diagnosis is still a challenge in Gem Sub-County, Kenya. We document that approximately one-third (31.3%) of diagnosed pediatric TB cases required three or up to 12 health facility visits before confirmation. This points to the severity of missed opportunities at the primary care level and provides actionable local evidence for Siaya County and similar decentralized health systems to initiate strategies to improve timely diagnosis. We demonstrate marked differences in diagnostic confidence and practice by HCW cadre, in particular, pediatric sample collection and chest X ray interpretation. A disparity not described in the Kenyan context before, hence need for practical guidance for targeted training interventions in resource-constrained settings where task-shifting is common. Further, we provide data showing that over half of caregivers explicitly linked childhood TB to HIV, and sought care only after persistent symptoms. These findings, illustrate how delayed recognition of pediatric TB symptoms in young children, stigma and low awareness interact to prolong diagnostic journeys in a rural African community.

This localized data is critical for informing county-level TB programming in Kenya's devolved health system and for similar resource-limited settings to adapt TB diagnosis and treatment guidelines to local realities. Future research is needed to explore influence of TB integration into maternal child health programs on TB screening and detection across full paediatric age spectrum (0–14 years) in particular, high-burden settings. Such integration could overcome current gaps in systematic screening and reduce missed opportunities in TB diagnosis.

## Acknowledgments

We thank the study participants for their participation in the study. We appreciate the Ministry of Health County Government of Siaya for the permission to conduct this study. We are grateful to the management of health facilities in Siaya, Kenya, for their support during the study.

## Author contributions

**Conceptualization:** Lilian Atieno Okumu.

**Investigation:** Lilian Atieno Okumu.

**Methodology:** Lilian Atieno Okumu.

**Writing – original draft:** Lilian Atieno Okumu.

**Writing – review & editing:** Daniel Ogungu Onguru, David Otieno Odongo, James Onyuro Oketch.

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
