## [Decision Letter · Decision Letter 0]

12 Nov 2025

Dear Dr. Okumu,

Thank you for submitting your manuscript to PLOS ONE. After careful consideration, we feel that it has merit but does not fully meet PLOS ONE’s publication criteria as it currently stands. Therefore, we invite you to submit a revised version of the manuscript that addresses the points raised during the review process.

Please submit your revised manuscript Dec 27 2025 11:59PM. If you will need significantly more time to complete your revisions, please reply to this message or contact the journal office at plosone@plos.org . A rebuttal letter that responds to each point raised by the academic editor and reviewer(s). You should upload this letter as a separate file labeled 'Response to Reviewers'.A marked-up copy of your manuscript that highlights changes made to the original version. You should upload this as a separate file labeled 'Revised Manuscript with Track Changes'.An unmarked version of your revised paper without tracked changes. You should upload this as a separate file labeled 'Manuscript'.

We look forward to receiving your revised manuscript.

Kind regards,

Frederick Quinn

Academic Editor

PLOS ONE

Additional Editor Comments (if provided):

Reviewers' comments:

Reviewer's Responses to Questions

**Comments to the Author**

1. Is the manuscript technically sound, and do the data support the conclusions?

Reviewer #1: Partly

Reviewer #2: Partly

2. Has the statistical analysis been performed appropriately and rigorously?

Reviewer #1: Yes

Reviewer #2: No

3. Have the authors made all data underlying the findings in their manuscript fully available?

Reviewer #1: Yes

Reviewer #2: No

4. Is the manuscript presented in an intelligible fashion and written in standard English?

Reviewer #1: Yes

Reviewer #2: Yes

Reviewer #1: The topic is highly relevant for public health and aligns with PLOS ONE’s scope. However, the findings mainly reinforce existing knowledge without offering strong new insights beyond local context. The authors should clarify what new contribution their study adds to the broader literature.

The sampling and data analysis sections are clearly explained but need stronger justification.

The sample size (16 caregivers, 71 HCWs) is small for meaningful regression analysis. The authors should explicitly acknowledge this limitation and avoid overinterpretation of odds ratios with extremely wide confidence intervals.

Multivariate analysis was omitted due to small sample size — this is appropriate but should be discussed as a limitation for causal inference.

Several odds ratios are very high (e.g., OR=34, 95% CI 3.81–303.21), indicating instability of estimates. This should be tempered in interpretation.

Tables should include p-values consistently and be formatted to avoid redundancy.

The qualitative quotes are rich and well-selected but should be more clearly integrated with quantitative findings in the discussion to illustrate triangulation of results.

The manuscript requires careful English editing to improve flow, avoid redundancy, and correct grammar (e.g., “pediatric population. This is mainly attributed…” should be merged into one sentence). Table formatting and figure captions should follow PLOS ONE guidelines.

Minor Comments:

• Clarify abbreviations at first use (e.g., “HCW,” “TB4 register”).

• Standardize referencing format per PLOS ONE style.

• Add clear subheadings in the Discussion (Diagnostic constraints, Patient factors, HCW factors).

• Strengthen the conclusion by focusing on actionable recommendations rather than restating results.

Reviewer #2: PONE-D-25-53931

Tuberculosis is a significant problem which plagues children, especially in low-and-middle-income countries. Identifying the factors that contribute to low tuberculosis diagnosis in children aged 0-14years in Gem County, Siaya County, Kenya, has the potential of contributing information that can help narrow, or even close the gap in the underdiagnosis of tuberculosis. Using a mixed method study is a laudable approach as it will provide in-depth exploration and better understanding of the subject matter. The authors have however not clearly stated the question (and objectives) clearly to guide the reader.

Use of non-standard symbols and syntax errors throughout the text are noted. The authors should be consistent with terminologies- use paediatric or child/children, Xpert MTB/RIF or GeneXpert. In- text references are required for facts being provided.

METHODOLOGY

The authors should elaborate more on the methodology used in this manuscript. A mixed study research, like any other research requires clarity. The authors should clearly define the mixed methods approach used and how it helped address their research question. They should describe the study design whether the study followed a convergent, explanatory

sequential, exploratory sequential, design and provide a rationale for their choice.

The sequence of data (quantitative and qualitative) collection should be explained. This will not only improve rigor, but also validity and reliability of the study.

It is not stated how the in-depth interviews were carried out among 71 health care workers and how this large volume of data was handled.

The semi structured questionnaire and in-depth interview questions have not been included in the manuscript.

RESULTS

This section needs to be structured to improve clarity in the mind of the readers and make it more meaningful. Mixed study requires that both qualitative and quantitative results are initially reported separately, not treated as separate entities but rather as complementary sources of evidence.

It is difficult to see where the two data sets were integrated.

The labelling of the tables and figures begin from 4.? There is no explanation for some of the figures in the body of the manuscript- Figures 4.2, 4.3, 4.4. Table 4.4, and 4.7 give no additional value to the manuscript.

DISCUSSION

The discussion starts with a repetition of the results. The authors need to integrate discussion that highlights areas of convergence, divergence, or complementarity of the data sets. In some places, cited articles are substitute by numbers with no context.

There should be a comment or two in the discussion section with regards to this statement- A 33-year-old link assistant noted, “Chest X-rays are not advisable in children due to harmful rays, which may lead to missed cases due to fear.”

The authors would have encountered several biases in this study (e.g. selection, social desirability, sampling, response etc)- How did the authors mitigate against biases?

The duration of the study (1- month) makes it imprudent to generalise the findings, as thus, should also be noted as a limitation. The factors influencing diagnosis might fluctuate seasonally or be influenced by specific short-term events (e.g. a temporary stock-out) that would not be representative of a typical situation throughout the year. Additionally, seasonal illnesses e.g. flu can mask TB symptoms and contribute to a delay in seeking care.

In general, the discussion section can be improved upon.

**Do you want your identity to be public for this peer review?** For information about this choice, including consent withdrawal, please see our Privacy Policy

Reviewer #1: No

Reviewer #2: No

---

## [Decision Letter · Decision Letter 1]

18 Jan 2026

Dear Dr. Okumu,

Thank you for submitting your manuscript to PLOS ONE. After careful consideration, we feel that it has merit but does not fully meet PLOS ONE’s publication criteria as it currently stands. Therefore, we invite you to submit a revised version of the manuscript that addresses the points raised during the review process.

Please submit your revised manuscript by Mar 04 2026 11:59PM. If you will need significantly more time to complete your revisions, please reply to this message or contact the journal office at plosone@plos.org . A letter that responds to each point raised by the academic editor and reviewer(s). You should upload this letter as a separate file labeled 'Response to Reviewers'.A marked-up copy of your manuscript that highlights changes made to the original version. You should upload this as a separate file labeled 'Revised Manuscript with Track Changes'.An unmarked version of your revised paper without tracked changes. You should upload this as a separate file labeled 'Manuscript'.

We look forward to receiving your revised manuscript.

Kind regards,

Frederick Quinn

Academic Editor

PLOS One

**Journal Requirements:**

Reviewers' comments:

Reviewer's Responses to Questions

**Comments to the Author**

Reviewer #2: (No Response)

2. Is the manuscript technically sound, and do the data support the conclusions?

Reviewer #2: Yes

3. Has the statistical analysis been performed appropriately and rigorously?

Reviewer #2: Yes

4. Have the authors made all data underlying the findings in their manuscript fully available?

Reviewer #2: Yes

5. Is the manuscript presented in an intelligible fashion and written in standard English?

Reviewer #2: Yes

Reviewer #2: Thank you for the opportunity to re-revise this manuscript.

The authors have addressed the corrections from the initial manuscript. The discussion section requires some rearrangement of some details in the appropriate sections to make for an easy read.

**Do you want your identity to be public for this peer review?** For information about this choice, including consent withdrawal, please see our Privacy Policy

Reviewer #2: No

You may also use PLOS’s free figure tool, NAAS, to help you prepare publication quality figures: https://journals.plos.org/plosone/s/figures#loc-tools-for-figure-preparation

---

## [Author Response · Author response to Decision Letter 2]

23 Jan 2026

Response to Reviewer comments

RESULTS

-(Line 186) There is need to specify the group of respondents being referred to. Half 9; 56.3% clearly does not refer to the HCW.

-The message conveyed in Fig 3 is unclear. The authors should tidy up the information on Tables 6 &7 OR summarize the information on them in prose form and expunge the tables.

Response: Specified, about half of HCWs were female (38, 53.5%), while over half of other respondents (9, 56.3%) were aged 35 years and above. Figure 3 has been removed from the manuscript as it conveyed limited additional value. Tables 6& 7 combined into one for clarity. Information presented in table 5

DISCUSSION

-Line 475-6: “CXR are not advisable for children…”- this is HCW misconception and should be moved to facility/HCW factors contributing to delayed diagnosis.

-Line 490-497: “Qualitative insights in skilled deficits…” should also be moved to facility/HCW factors contributing to delayed diagnosis.

-Line 563 “The absence of essential tools, coupled with poor sample collection techniques, was a significant constraint to TB diagnosis” should come under diagnostic constraints.

-Line 558- The sentence here is a repetition as it speaks to HCW technique and should be moved the appropriate section

Response: Line 475 moved to facility/HCW factors contributing to delayed diagnosis.

Line 490 (qualitative insights) moved as appropriate

Line 563 concern is acknowledged. However, considered as facility/HCW factors

Line 558, moved to appropriate section

---

## [Decision Letter · Decision Letter 2]

4 Feb 2026

Factors Contributing to Low Tuberculosis Diagnosis Among Children Aged 0-14 Years in Gem Sub County in Siaya County, Kenya.

PONE-D-25-53931R2

Dear Dr. Okumu,

We’re pleased to inform you that your manuscript has been judged scientifically suitable for publication and will be formally accepted for publication once it meets all outstanding technical requirements.

Kind regards,

Frederick Quinn

Academic Editor

PLOS One

Additional Editor Comments (optional):

Reviewers' comments:

Reviewer's Responses to Questions

**Comments to the Author**

Reviewer #2: All comments have been addressed

2. Is the manuscript technically sound, and do the data support the conclusions?

Reviewer #2: Yes

3. Has the statistical analysis been performed appropriately and rigorously?

Reviewer #2: Yes

4. Have the authors made all data underlying the findings in their manuscript fully available?

Reviewer #2: Yes

5. Is the manuscript presented in an intelligible fashion and written in standard English?

Reviewer #2: Yes

Reviewer #2: Thank you for the opportunity to revise the manuscript titled- Factors Contributing to Low Tuberculosis Diagnosis Among Children Aged 0-14 Years in Gem Sub County in Siaya County, Kenya. The authors have made the corrections in all the areas highlighted. I recommend that the manuscript be accepted for publication.

**Do you want your identity to be public for this peer review?** For information about this choice, including consent withdrawal, please see our Privacy Policy

Reviewer #2: No

---

## [Editor Report · Acceptance letter]

PONE-D-25-53931R2

PLOS One

Dear Dr. Okumu,

I'm pleased to inform you that your manuscript has been deemed suitable for publication in PLOS One. Congratulations! Your manuscript is now being handed over to our production team.

Kind regards,

on behalf of

Dr. Frederick Quinn

Academic Editor

PLOS One